# Level of cognitive functioning among elderly patients in urban area of Bangladesh: A cross-sectional study

**Joynal Abedin Imran**[1,2☯] *, **Pradip Kumar Saha**[1,2☯], **Marzana Afrooj Ria**[1,2☯], **Syeda Saika Sarwar**[1‡], **Jannatul Ferdous Konok**[1‡]

**1** Department of Physiotherapy, National Institute of Traumatology & Orthopaedic Rehabilitation (NITOR), Dhaka, Bangladesh, **2** Bangladesh Gerontological Foundation, Bangladesh

☯ These authors contributed equally to this work.
‡ SSS and JFK also contributed equally to this work.
* 4abedin@gmail.com, j.abedin@traumaimt.edu.bd

**Data Availability Statement:** All relevant data are within the manuscript and its Supporting information files.

## Abstract

Bangladesh is experiencing rapid urbanization and a growing elderly population, particularly in urban areas. Cognitive decline, ranging from mild cognitive impairment to dementia, is a prevalent issue among elderly populations globally. Understanding the current state of cognitive functioning in this demographic is essential for informing effective healthcare plans and programs. This study aims to investigate the prevalence of cognitive decline and its associated factors among urban-dwelling elderly adults in Bangladesh, using the Rowland Universal Dementia Assessment Scale (RUDAS) to assess cognitive function. This cross-sectional study employed systematic random sampling among 150 elderly participants (aged 60–85 years) from the outpatient department of the National Institute of Traumatology and Orthopaedic Rehabilitation (NITOR) in Dhaka, Bangladesh. The mean age of participants was 67.41 ± 6.31 years, with a male predominance (53.3%). Cognitive function was impaired in a majority of participants, with 53.3% classified as having dementia, 38.7% with MNCD, and only 8% showing normal cognitive function. Significant predictors of cognitive function included age (r = -0.451, P < 0.001), educational level (P = 0.009), and diabetes (P = 0.038). Female participants had lower mean cognitive function scores compared to males (21.16 ± 5.25 vs. 22.03 ± 4.36, P = 0.271). Cognitive impairment is highly prevalent among elderly individuals in urban Bangladesh, with age, educational level, and diabetes being key predictors. These findings highlight the need for public health interventions and policies focused on early screening and targeted healthcare for cognitive decline in this demographic.

## Introduction

Cognitive decline is a prevalent issue among elderly populations globally [1], and Bangladesh is not exempt from this trend. The nation, which is undergoing swift urbanization, is witnessing a rise in its elderly population, especially in urban regions [2]. The swift urbanization and

**Funding:** The author(s) received no specific funding for this work.

**Competing interests:** The authors have declared that no competing interests exist.

the aging population in Bangladesh have heightened concerns about the mental health and cognitive abilities of the elderly. This demographic change presents substantial challenges as well as opportunities for healthcare systems.

Cognitive impairment, which can lead to dementia, is a primary cause of disability among the elderly and a leading factor in dependency for older adults across low-, middle-, and high-income countries [3]. This impairment is a significant public health issue as it greatly affects an individual's quality of life, independence, and daily functioning. Mild cognitive impairment (MCI) represents an intermediate stage between normal cognitive function and dementia [4]. Research indicates that individuals with MCI are at a higher risk of developing dementia Currently, dementia impacts around 10 million people in the region, with its prevalence expected to double by 2030 [5]. Globally, approximately 35.6 million people suffer from dementia, a number projected to double to 65.7 million by 2030 and more than triple to 115.4 million by 2050. Dementia affects individuals worldwide, with over half (58%) residing in low- and middle-income countries; by 2050, this proportion is expected to exceed 70% In Bangladesh, Alzheimer's and dementia-related deaths accounted for 14,993, or 2.09% of all deaths, according to the latest WHO data published in 2020 [6].

Currently, the majority of Bangladesh's population is young; however, in the next 20 to 30 years, the proportion of older individuals is expected to increase significantly [7]. The urban elderly population in Bangladesh is growing rapidly, making it crucial to understand their cognitive functioning in order to develop effective healthcare strategies and interventions. There is a notable absence of government policies directly addressing the burden of cognitive impairment or dementia, largely due to the lack of reliable research that could inform policymakers, health professionals, and relevant authorities. Additionally, there is very limited research on cognitive impairment among the elderly in Bangladesh, which has not garnered sufficient attention from the research community. This study aims to address this gap by estimating the current prevalence of cognitive impairment and investigating its potential predictors among older adults in urban area of Bangladesh.

## Materials and methods

The research employed an observational method featuring quantitative data within a cross-sectional study design. Data were collected from elderly patients visiting the outpatient department at National Institute of Traumatology & Orthopaedic Rehabilitation (NITOR) in Dhaka during 3rd March to 11th June 2024. While a population-based study would provide a more comprehensive assessment of cognitive decline prevalence in the community, the current study utilizes a hospital-based sample at the National Institute of Traumatology and Orthopaedic Rehabilitation (NITOR) to capture a segment of the urban elderly population who have access to healthcare services. This approach was selected due to logistical and resource constraints, as well as the ease of reaching elderly participants at a single, centralized location where patients with potential health issues, including cognitive impairment, frequently seek medical care. As such, the study findings provide valuable insights into cognitive decline among elderly patients in an urban setting, although future research could expand to a population-based design for broader generalizability. Inclusion criteria was set required to be aged 60 years or older, reside in urban Dhaka, and visit the outpatient department at NITOR during the study period. Both male and female participants were included, provided they were able to give informed consent and complete the cognitive assessment. Exclusion Criteria were set as follows, individuals with a previous diagnosis of dementia or other severe neurological disorders that could interfere with cognitive function were excluded. Additionally, those with severe

sensory impairments (e.g., profound hearing or vision loss) or psychiatric conditions that could prevent them from completing the assessment were not eligible for the study.

The sample size was determined based on an estimated prevalence rate of cognitive impairment in elderly populations in South Asia, which generally ranges between 10–15%. We selected a conservative estimate of 10% [8] as our prevalence rate to ensure adequate power for detecting cognitive impairment among urban elderly individuals in Bangladesh. The following formula was used for the calculation:

$$n = \frac{Z^2 p(1-p)}{d^2}$$

Where:

Z = 1.96 for a 95% confidence level,

p = 0.10 as the estimated prevalence of cognitive impairment, and

d = 0.05 representing the margin of error.

This formula was selected to ensure adequate statistical power and precision in estimating cognitive impairment prevalence in the study population. Based on these parameters, a minimum sample size of 150 participants was determined as sufficient for capturing meaningful prevalence and associations of cognitive impairment among elderly individuals in an urban Bangladeshi setting. Informants were chosen using a systematic random sampling technique. The sampling interval was determined using the following formula:

$$K = \frac{N}{n}$$

$$K = \frac{476.03}{150}$$

$$K = 3.17$$

Here, N = Number of Population (173751 patients visited outpatient department, NITOR annually at 2022 [9] according to this data 476.03 patients visited daily), n = Number of sample size (150), K = Interval size.

Following the sampling interval, every third elderly individual was approached and invited to participate in an interview. Prior to participation, each selected individual was required to provide written informed consent. In instances where a participant declined to participate, the next eligible individual, subsequent to every third person, was approached and invited to participate. The study was conducted in accordance with the declaration of Helsinki, and the protocol was approved by ethics committee of institutional review board of NITOR (Reference no. NITOR/PT/93/IRB/2024/4). Prior to commencing the study, participants were comprehensively apprised of the research objectives and methodologies. They were explicitly informed that their participation was entirely voluntary and that they could withdraw at any time. Furthermore, participants were assured that their data would be treated with the utmost confidentiality and anonymity, and that they would be fully informed of any potential risks and benefits associated with their involvement. Notably, no physical specimens were collected during the course of the study. Rather, researchers employed a standardized questionnaire to collect data, which is detailed below.

## Questionnaire

Elderly patients were interviewed using a structured questionnaire to gather information on sociodemographic characteristics and physical health status. The sociodemographic characteristics recorded for each participant included age, gender, marital status, educational level, occupation, and monthly family income. These variables were chosen to analyze potential associations with cognitive function and to provide a comprehensive profile of the study population. Regarding physical health status, data were collected on body mass index (BMI), smoking status, daily sleep duration, and the presence of comorbidities such as diabetes, hypertension, stroke history, vision impairment, arthritis, and other chronic conditions. These factors were considered for their potential impact on cognitive function and were analyzed to identify any correlations with cognitive impairment. Cognitive function was assessed using the Rowland Universal Dementia Assessment Scale (RUDAS), a standard tool developed in a multicultural context in Australia [10]. This brief, six-item test is designed for quick and simple administration, is portable, and is considered culturally and educationally unbiased. The RUDAS evaluates body orientation, praxis, drawing, judgment, memory, and language, and also assesses executive function impairment. Scores were calculated according to the RUDAS scoring manual. According to Basic [10], a score below 23 points indicates possible dementia, prompting referral for further neuropsychological evaluation. Scores above 27 points are considered normal, while scores between 23 and 27 points may suggest a mild neurocognitive disorder (MNCD). In our study, the survey instrument was translated into Bengali using forward and backward translation and field-tested among 30 elderly participants. Any inconsistencies of data were identified according to necessity and corrected during field level. Regarding content validity after translation process for RUDAS scale, Cronbach's Alpha value was 0.9 and Cronbach's Alpha based on standardized items was 0.906 indicated that the instrument has an acceptable level of validity.

## Statistical analysis

The data analysis was performed using IBM SPSS version 26. Initially, descriptive statistical analysis was conducted to understand the participants' characteristics, including their RUDAS scores. Pearson's chi-square test or Fisher's exact test, students t-test was utilized to explore the association between cognitive function and sociodemographic factors, physical health, and comorbidities. Bivariate correlations between cognitive function and continuous factors were assessed using correlation coefficients and categorical factors were assess using t-test and ANOVA to identify significant relationships. For multivariate analysis, a linear regression model was developed with the RUDAS score as the dependent variable. Only covariates significantly associated with cognitive function in the bivariate analyses were included in the multivariate model. This approach controlled for potential confounding effects and identified the independent effects of each factor on cognitive function. Statistical significance was set at a P-value of less than 0.05. This comprehensive statistical analysis approach allowed for an in-depth examination of the relationship between sociodemographic factors and cognitive function, controlling for confounding effects and identifying independent predictors of cognitive function in the elderly.

## Results

### Sociodemographic characteristics

In this study, 150 adults were enrolled, and none of the participants dropped out. The participants' ages ranged from 60 to 85 years, with a mean age of 67.41 ± 6.31 years (95% confidence

**Table 1. Sociodemographic characteristics of the participants (n = 150).**

| Characteristics | All participants (*n* = 150) | Male (*n* = 80) | Female (*n* = 70) | Chi-square | *P*—value |
|---|---|---|---|---|---|
| **Age (mean ± *SD*)** | 67.41 ± 6.31 | 67.50 ± 6.37 | 67.31 ± 6.30 | - | 0.874[a] |
| **Marital status (%)** | | | | 43.096[b] | <0.001 |
| Married | 118 (78.7) | 74 (92.5) | 44 (62.9) | | |
| Unmarried | 2 (1.3) | 1 (1.3) | 1 (1.4) | | |
| Widow/Widower | 30 (20) | 5 (6.3) | 25 (35.7) | | |
| **Education (%)** | | | | 35.706[c] | <0.001 |
| No formal education | 17 (11.3) | 4 (5) | 13 (18.6) | | |
| Primary education (grade 1–5) | 42 (28) | 11 (13.8) | 31 (44.3) | | |
| Secondary education (grade 6–10) | 23 (15.3) | 13 (16.3) | 10 (14.7) | | |
| Higher secondary (grade 11–12) | 33 (22) | 22 (27.5) | 11 (15.7) | | |
| Graduation | 22 (14.7) | 19 (23.8) | 3 (4.3) | | |
| Post-graduation | 13 (8.7) | 11 (13.8) | 2 (2.9) | | |
| **Occupation (%)** | | | | 148.568[b] | <0.001 |
| Retired | 32 (21.3) | 28 (35) | 4 (5.7) | | |
| Farmer | 11 (7.3) | 11 (13.8) | 0 | | |
| Businessman | 22 (14.7) | 22 (27.5) | 0 | | |
| Service holder | 12 (8) | 10 (12.5) | 2 (2.9) | | |
| Unemployed | 5 (3.3) | 5 (6.3) | 0 | | |
| Housewife | 61 (40.7) | 0 | 61 (87.1) | | |
| Others | 7 (4.7) | 4 (5) | 3 (4.3) | | |
| **Family monthly expense (Taka)[d]** | | | | 3.886[c] | 0.274 |
| Below 20,000 | 29 (19.3) | 12 (15) | 17 (24.3) | | |
| 20,000–39,999 | 74 (49.3) | 39 (48.8) | 35 (50) | | |
| 40,000–59,999 | 35 (23.3) | 23 (28.7) | 12 (17.1) | | |
| 60,000 and above | 12 (8) | 6 (7.5) | 6 (8.6) | | |

[a]t-test; 95% confidence intervals (CI) for mean difference = -1.76 to 2.08;

[b]Fisher's exact correction test used for having 20% or more expected frequencies less than 5;

[c]Pearson Chi-square;

[d]1 US dollar = 119.12 Taka in November, 2024.

intervals = -1.76 to 2.08). The majority of participants were male (53.3%, n = 80), married (78.7%, n = 118), and had primary education (28%, n = 42) (Table 1). Regarding monthly family income, 68.6% (n = 103) were classified as low-income families. The mean age of males was slightly higher than that of females. Additionally, there were gender differences in marital status, education levels, and occupation.

## Differences in characteristics of physical health and comorbidities related factors

Table 2 presents characteristics related to the physical health, smoking habits, sleep patterns, comorbidities, and cognitive function of the elderly population. The participants had a normal BMI, with a mean ± SD of 23.99 ± 4.06. The obesity rate was slightly higher in males (7.5%) compared to females (7.1%). Due to social norms in Bangladesh, none of the female participants smoked, while 15% of the male participants were current or former smokers. Most elderly individuals (54.7%) reported getting 6 to 7 hours of sleep per day. Diabetes, stroke, and

**Table 2. Characteristics of physical health, sleep hours, comorbidities and RUDAS score of the participants (n = 150).**

| Characteristics | All participants (*n* = 150) | Male (*n* = 80) | Female (*n* = 70) | Chi-square | *P*—value |
|---|---|---|---|---|---|
| **Body mass index**, (mean ± SD) | 23.99 ± 4.06 | 24.50 ± 4.25 | 23.41 ± 3.78 | | 0.102[a] |
| Underweight | 3 (2) | 0 | 3 (4.3) | 3.611 | 0.307[b] |
| Normal | 106 (70.7) | 56 (70) | 50 (71.4) | | |
| Overweight | 30 (20) | 18 (22.5) | 12 (17.1) | | |
| Obese | 11 (7.3) | 6 (7.5) | 5 (7.1) | | |
| **Smoking status (%)** | | | | 61.765 | <0.001[c] |
| Current smoker | 12 (8) | 12 (15) | 0 | | |
| Former Smoker | 36 (24) | 36 (45) | 0 | | |
| Never smoked | 102 (68) | 32 (40) | 70 (100) | | |
| **Water intake**, (mean ± SD) | 7.50 ± 1.84 | 7.89 ± 1.96 | 7.06 ± 1.59 | - | 0.005[a] |
| **Sleep duration per day** | 6.72 ± 1.25 | 6.81 ± 1.23 | 6.61 ± 1.28 | - | 0.339[a] |
| 3–5 hours | 24 (16) | 9 (11.3) | 15 (21.4) | | |
| 6–7 hours | 82 (54.7) | 48 (60) | 34 (48.6) | | |
| 8–9 hours | 44 (29.3) | 23 (28.7) | 21 (30) | | |
| **Diabetes (%)** | | | | 1.467 | 0.250[c] |
| Yes | 65 (43.3) | 31 (38.8) | 34 (48.6) | | |
| No | 85 (56.7) | 49 (61.3) | 36 (51.4) | | |
| **Stroke (%)** | | | | 3.022 | 0.082[c] |
| Yes | 33 (22) | 22 (27.5) | 11 (15.7) | | |
| No | 117 (78) | 58 (72.5) | 59 (84.3) | | |
| **Cardiac disease (%)** | | | | 2.406 | 0.121[c] |
| Yes | 57 (38) | 35 (43.8) | 22 (31.4) | | |
| No | 93 (62) | 45 (56.3) | 48 (68.6) | | |
| **Eye sight problem (%)** | | | | 6.780 | <0.01[b] |
| Yes | 95 (63.3) | 43 (53.8) | 52 (74.3) | | |
| No | 55 (36.7) | 37 (46.3) | 18 (25.7) | | |
| **Arthritis (%)** | | | | 5.788c | <0.05[c] |
| Yes | 108 (72) | 51 (63.7) | 57 (81.4) | | |
| No | 42 (28) | 29 (36.3) | 13 (18.6) | | |
| **Pulmonary disease (%)** | | | | 0.010 | 0.920[c] |
| Yes | 38 (25.3) | 20 (25) | 18 (25.7) | | |
| No | 112 (74.7) | 60 (75) | 52 (74.3) | | |
| **RUDAS**, (mean ± SD) | 21.62 ± 4.80 | 22.03 ± 4.36 | 21.16 ± 5.25 | - | 0.271[a] |
| Memory | 5.47 ± 1.89 | 5.50 ± 1.92 | 5.43 ± 1.86 | - | 0.818[a] |
| Visuosaptial orientation | 4.67 ± 0.70 | 4.75 ± 0.66 | 4.59 ± 0.73 | - | 0.152[a] |
| Praxis | 1.03 ± 0.51 | 1.09 ± 0.45 | 0.97 ± 0.56 | - | 0.172[a] |
| Visuoconstructional Drawing | 1.58 ± 1.14 | 1.65 ± 1.08 | 1.50 ± 1.29 | - | 0.427[a] |
| Judgement | 3.03 ± 1.17 | 3.05 ± 1.23 | 3.01 ± 1.11 | - | 0.852[a] |
| Language | 5.83 ± 1.30 | 5.99 ± 1.32 | 5.66 ± 1.27 | - | 0.122[a] |
| **RUDAS score categories (%)** | | | | 0.489 | 0.783[c] |
| Dementia (less than 23) | 80 (53.3) | 41 (51.2) | 39 (55.7) | | |
| Mild Neurocognitive Disorder (MNCD) (23–27) | 58 (38.7) | 33 (41.3) | 25 (35.7) | | |
| Normal (Above 27) | 12 (8) | 6 (7.5) | 6 (8.6) | | |

[a]t-test,

[b]Fisher's exact correction test used for having 20% or more expected frequencies less than 5;

[c]Pearson Chi-Square.

cardiac disease were more prevalent among males, whereas sleeping difficulties, vision problems, arthritis, and asthma were more common among females.

**RUDAS scores.**   The study participants (n = 150) generally exhibited poor cognitive function, with a mean ± SD score of 21.62 ± 4.80. Females had lower cognitive function scores compared to males (21.16 ± 5.25 vs. 22.03 ± 4.36). When categorizing cognitive function, the majority of participants (53.3%) were classified as having poor cognitive function (Dementia), over a quarter had mild neurocognitive disorder (MNCD), and only about 8% had normal cognitive function.

## Distribution of elderly people by cognitive function score (RUDAS) categories

Among the 150 participants, the highest cognitive function score was 29 and the lowest was 8, out of a possible 30 points. The median cognitive function score was 22, with the 25th and 75th percentiles being 19 and 25, respectively. A large portion of the participants (53.3%) were identified as having dementia, while only 8% were observed to have normal cognitive function (Fig 1).

## Correlation between sociodemographic factors and cognitive function (RUDAS) scores

A bivariate correlation test (Table 3) revealed that cognitive function was inversely correlated with age (r = -0.451, P = 0.000), indicating poorer cognitive function with increasing age among the elderly population.

A bivariate t-test and ANOVA test that revealed Education (P = 0.009), Diabetes (P = 0.038), and Eye Sight Problems (P = 0.009) are significantly associated with RUDAS

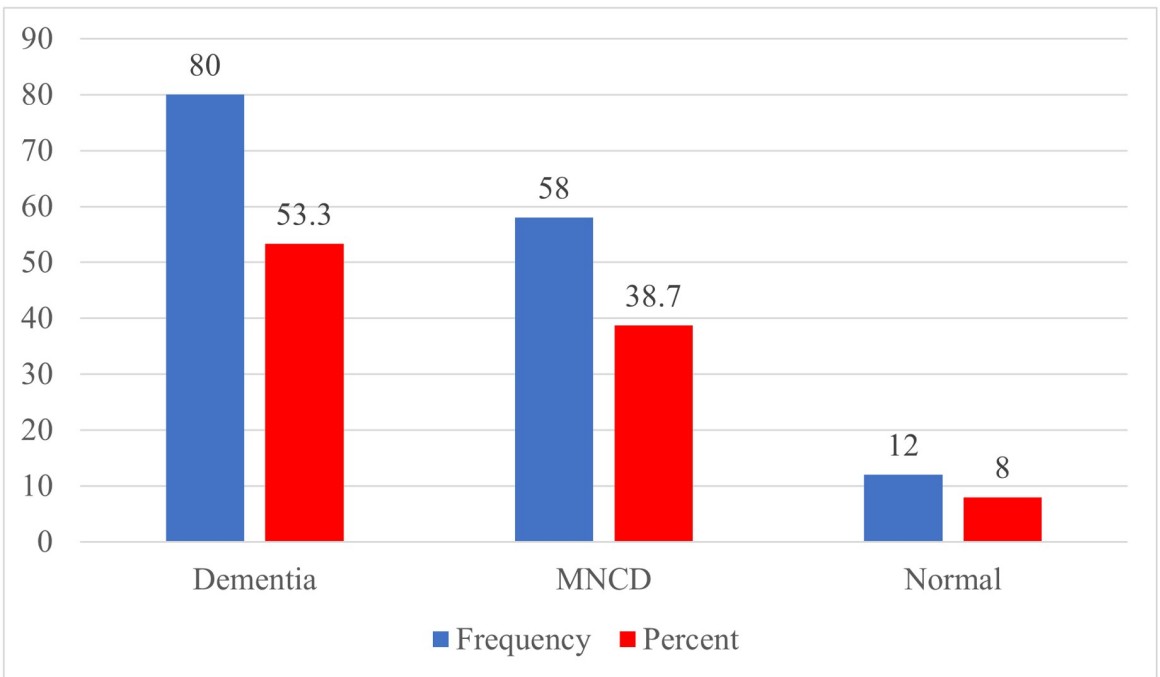

**Fig 1. Distribution of the participants by RUDAS scores (n = 150).**

**Table 3. Correlation matrix between RUDAS scores with sociodemographic and comorbidities factors in elderly population (n = 150).**

|  |  | Age | Monthly family expense | BMI | Water Intake | Sleep duration | RUDAS |
|---|---|---|---|---|---|---|---|
| Age | r | 1 | 0.073 | 0.078 | -0.033 | 0.085 | -0.451** |
|  | P |  | 0.375 | 0.341 | 0.690 | 0.303 | 0.000 |
| Monthly family expense | R |  | 1 | 0.094 | -0.049 | -0.062 | 0.109 |
|  | P |  |  | 0.251 | 0.548 | 0.453 | 0.184 |
| BMI | r |  |  | 1 | -0.080 | 0.163* | 0.138 |
|  | P |  |  |  | 0.331 | 0.047 | 0.092 |
| Water intake | r |  |  |  | 1 | 0.020 | 0.104 |
|  | P |  |  |  |  | 0.806 | 0.207 |
| Sleep duration | r |  |  |  |  | 1 | 0.039 |
|  | P |  |  |  |  |  | 0.637 |

**Correlation is significant at the 0.01 level (2-tailed);

*Correlation is significant at the 0.05 level (2-tailed)

Scores implying they were significant predictors understanding cognitive performance (Table 4).

## Linear regression analysis to predict cognitive function

The variables that were statistically significant in the bivariate analysis were included in the regression model. The dependent variable was the continuous cognitive function (RUDAS) score. The selected independent variables were: age, educational status (0 = graduation & below, 1 = post-graduation), diabetes (0 = yes, 1 = no), and eyesight problems (0 = yes, 1 = no). Before developing the model multicollinearity and residual assumptions were analyzed. No multicollinearity was observed among the independent variables. The standard residual value was minimum -2.896 and maximum 2.357 indicating no outliers in the dependent variable. The Durbin-Watson test value was 1.276 indicating data points were independent. To analyze the normality of the residuals and constant variance; histogram, normal probability (P-P) plot and a scatter plot of the residuals were produced. The distribution of the residuals was seen normal in the histogram and P-P plot. The variance (homoscedasticity) was constant in the scatter plot.

Table 5 data shows that the statistically significant predictors of cognitive function including: age ($P = <0.001$), educational status ($P = 0.046$), history of diabetes ($P = 0.005$). Eye sight problem was excluded from the model because of $P > 0.05$.

The predicted model for the data was as follows:

$$\text{Cognitive function(RUDAS)} = 41.96 - 0.33(\text{age}) + 2.4(\text{educational status}) + 1.95(\text{history of diabetes})$$

The multiple linear regression analysis identified three significant predictors of cognitive function among elderly patients: education, age, and diabetes. These predictors collectively explained a significant proportion of the variance in cognitive function scores. Regarding education, Higher levels of education were associated with better cognitive function ($\beta = +2.4$, $P < 0.046$). This finding highlights the protective role of education in maintaining cognitive abilities in older adults. Advancing age was negatively associated with cognitive function ($\beta = -0.33$, $P < 0.001$). This suggests that aging is a significant risk factor for cognitive decline.

**Table 4. Association between RUDAS scores with sociodemographic and comorbidities factors in elderly population (n = 150).**

| Variables | RUDAS Score | | P-value |
|---|---|---|---|
| | Mean | SD | |
| **Gender** | | | 0.271[a] |
| Male | 22.03 | 4.36 | |
| Female | 21.16 | 5.25 | |
| **Marital Status** | | | 0.528[b] |
| Married | 21.85 | 4.58 | |
| Unmarried | 21.50 | 6.36 | |
| Widow/widower | 20.73 | 5.59 | |
| **Education** | | | 0.009[a] |
| Graduation & below | 21.34 | 4.81 | |
| Post graduation | 24.62 | 3.66 | |
| **Occupation** | | | 0.347[b] |
| Retired | 22.03 | 4.43 | |
| Farmer | 20.36 | 5.03 | |
| Businessman | 22.23 | 3.68 | |
| Service holder | 24.08 | 3.50 | |
| Unemployed | 19.20 | 5.81 | |
| Housewife | 21.05 | 5.36 | |
| Others | 22.29 | 4.96 | |
| **Smoking status** | | | 0.636[b] |
| Current smoker | 21.83 | 3.79 | |
| Former smoker | 22.25 | 4.14 | |
| Never smoked | 21.37 | 5.13 | |
| **Diabetes** | | | 0.038[a] |
| Present | 20.68 | 5.04 | |
| Absent | 22.34 | 4.52 | |
| **Stroke** | | | 0.446[a] |
| Present | 20.97 | 5.74 | |
| Absent | 21.80 | 4.52 | |
| **Cardiac disease** | | | 0.606[a] |
| Present | 21.37 | 4.44 | |
| Absent | 21.77 | 5.04 | |
| **Eye sight problem** | | | 0.009[a] |
| Present | 20.84 | 5.16 | |
| Absent | 22.96 | 3.81 | |
| **Arthritis** | | | 0.292[a] |
| Present | 21.37 | 4.91 | |
| Absent | 22.26 | 4.51 | |
| **Pulmonary disease** | | | 0.919[a] |
| Present | 21.68 | 4.32 | |
| Absent | 21.60 | 4.98 | |

[a]t-test;
[b]ANOVA

**Table 5. Factors predicting cognitive function in elderly people (n = 150).**

| Variable | β- coefficient | SE | Standardized β- coefficient | P-value | 95% CI for β |
|---|---|---|---|---|---|
| Constant | 41.96 | 3.74 | | <0.001 | |
| Age | -0.33 | 0.06 | -0.43 | <0.001 | -0.43 to -0.22 |
| Educational status | 2.4 | 1.20 | 0.14 | 0.046 | 0.05 to 4.79 |
| Diabetes | 1.95 | 0.68 | 0.20 | 0.005 | 0.60 to 3.30 |

Variables excluded from the model: Eye sight problem (P = 0.058)

Diabetes was found to adversely impact cognitive function (β = -1.95, P < 0.005), underscoring its role as a modifiable risk factor. See Fig 2.

## Discussion

Findings based on the RUDAS scale in this study indicate that a significant number of participants aged 60 years and older exhibited mild neurocognitive impairment and dementia. Similar results have been reported in several studies, which found that cognitive performance significantly decreases with increasing age [11–13]. Additionally, our study found that females had more impaired cognitive function than males, a trend also noted by Beam et al. and Oishee et al. [12, 14].

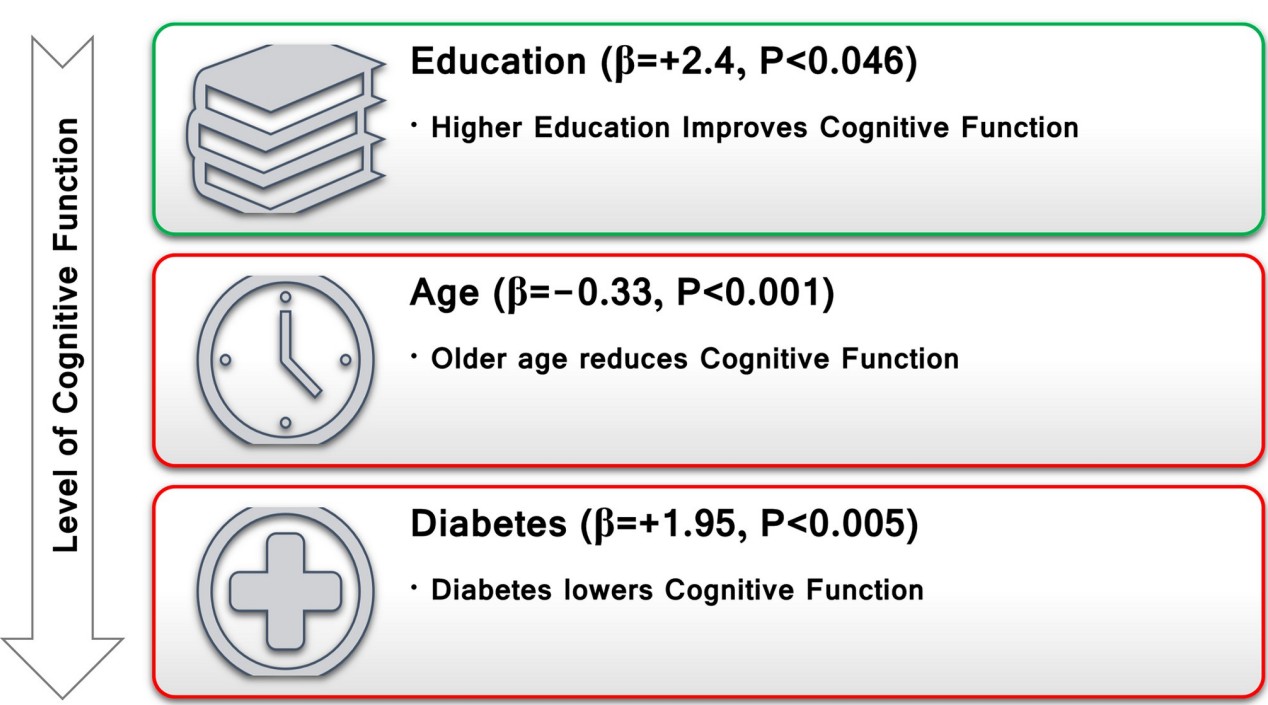

**Fig 2. Summary of significant predictors influencing cognitive function among elderly patients in an urban area of Bangladesh.**

In this study, educational qualifications of the elderly showed a significant relationship with cognitive function. Higher educational status was associated with better cognitive function in the elderly. Research supports that higher education levels correlate with improved cognitive health and can serve as a protective factor against cognitive decline [15, 16]. Additionally, the level of education moderates the relationship between depressive symptoms and cognitive functioning, with lower education levels being linked to poor cognitive outcomes [17].

This study found a higher prevalence of diabetes among female participants, which was significantly associated with cognitive decline. Diabetic elderly participants had lower cognitive function compared to non-diabetic participants. Elderly individuals with diabetes are at increased risk for cognitive dysfunction and dementia, affecting their overall cognitive functioning. Research supports that diabetes, whether diagnosed or undiagnosed, is linked to poor cognitive function, particularly in the elderly [18, 19]. Cognitive impairment among those aged 65 and older, as measured by the Mini-Mental State Examination (MMSE), was significantly higher among diabetic patients [20].

Although this study had found no significant relationship between eyesight problems and cognitive function but it is important to consider the possibility of a bidirectional relationship, especially in cases of vascular dementia. Cognitive decline, particularly due to vascular pathology, may impair visual processing abilities, potentially leading to reduced visual acuity [21]. Some study supports a strong association between cognitive function and visual ability in the elderly. Studies have shown that individuals with mild cognitive impairment (MCI) have lower functional visual acuity compared to those with normal cognition [22], and worsening vision is linked to declining cognitive function over time [23]. Improved cognitive function is associated with better best-corrected visual acuity, highlighting the importance of addressing visual impairments to mitigate cognitive decline in older adults [24].

To fully understand the findings of this study, it's essential to take into account the cultural nuances and factors that are specific to Bangladesh, as they may have a significant impact on the cognitive health of elderly individuals in that country. In Bangladesh, older generations typically have lower levels of formal education, a factor consistently linked to increased risk for cognitive decline [25]. Socioeconomic challenges also play a significant role, as access to healthcare resources, particularly for mental and preventive health, is limited, contributing to potentially unaddressed or worsening cognitive impairment [26]. Physical activity, another crucial factor for cognitive health, tends to be lower among elderly individuals in urban areas, impacting overall cognitive outcomes [27]. Social norms in Bangladesh, including the strong cultural expectation for family members to provide care for elderly relatives, may further influence the detection and management of cognitive impairment. This reliance on family caregiving may delay formal healthcare interventions, as families often prioritize home-based care until cognitive symptoms become severe [28]. These sociocultural factors underscore the need for targeted health interventions that consider both the medical and social needs of elderly populations in Bangladesh. Tailoring public health initiatives to address these specific cultural influences may improve both the prevention and management of cognitive impairment within this demographic.

## Limitations

This study was conducted in a specific area of Bangladesh, chosen purposively due to the cooperation of local authorities. To address these limitations, this study is the lack of a specific assessment for depressive symptoms, which could contribute to cognitive impairment in a phenomenon often referred to as "pseudo-dementia." Depression in elderly individuals can mimic symptoms of cognitive decline, complicating accurate differentiation between true cognitive

impairment and cognitive symptoms due to depression. This distinction is particularly relevant given the prevalence of underdiagnosed depression in older adults and its potential impact on cognitive assessments. Moving forward, future studies should explore additional factors that may influence cognitive function and incorporate longitudinal designs to better understand the trajectory of cognitive decline over time as this study provided a snapshot of an elderly population at a specific moment, making it difficult to assess temporal relationships or trends.

## Conclusion

Efforts to develop tailored interventions and policies should consider the multifaceted nature of cognitive health and the diverse needs of aging populations in Bangladesh and beyond. By addressing these predictors and implementing targeted interventions, Government & NGO's can work towards improving cognitive function and quality of life for elderly individuals in community level at Bangladesh.

## Supporting information

**S1 Appendix. Informed consent form.** The included written consent form administered to the participants.
(DOCX)

**S2 Appendix. Questionnaire.** The included questions comprised the questionnaire administered to the participants.
(DOCX)

**S3 Appendix. The included data of the participants.**
(CSV)

## Acknowledgments

We would like to recognize the efforts of Sanjib Saha, Associate researcher, Lund University, Sweden who helped us through this study.

## Author Contributions

**Conceptualization:** Joynal Abedin Imran, Pradip Kumar Saha.

**Data curation:** Syeda Saika Sarwar.

**Formal analysis:** Joynal Abedin Imran.

**Investigation:** Marzana Afrooj Ria, Syeda Saika Sarwar.

**Methodology:** Joynal Abedin Imran.

**Project administration:** Pradip Kumar Saha.

**Resources:** Marzana Afrooj Ria, Jannatul Ferdous Konok.

**Software:** Joynal Abedin Imran.

**Supervision:** Pradip Kumar Saha.

**Validation:** Pradip Kumar Saha.

**Visualization:** Joynal Abedin Imran, Pradip Kumar Saha.

**Writing – original draft:** Joynal Abedin Imran.

**Writing – review & editing:** Joynal Abedin Imran, Marzana Afrooj Ria, Jannatul Ferdous Konok.

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
