## [Decision Letter · Decision Letter 0]

15 Oct 2024

PONE-D-24-35131Level of Cognitive Functioning among Elderly Patients in Urban area of Bangladesh: A Cross-Sectional StudyPLOS ONE

Dear Dr. Abedin,

Thank you for submitting your manuscript to PLOS ONE. After careful consideration, we feel that it has merit but does not fully meet PLOS ONE’s publication criteria as it currently stands. Therefore, we invite you to submit a revised version of the manuscript that addresses the points raised during the review process.

The reviewers found some vital issues in the study design. I'd like to see the responses you present for the reviewers' comments.

We look forward to receiving your revised manuscript.

Kind regards,

Kallol Kumar Bhattacharyya, MBBS MA PhD

Academic Editor

PLOS ONE

Journal Requirements:

Reviewers' comments:

Reviewer's Responses to Questions

**Comments to the Author**

1. Is the manuscript technically sound, and do the data support the conclusions?

Reviewer #1: No

Reviewer #2: Yes

Reviewer #3: Yes

2. Has the statistical analysis been performed appropriately and rigorously? 

Reviewer #1: No

Reviewer #2: Yes

Reviewer #3: I Don't Know

3. Have the authors made all data underlying the findings in their manuscript fully available?

Reviewer #1: No

Reviewer #2: Yes

Reviewer #3: Yes

4. Is the manuscript presented in an intelligible fashion and written in standard English?

Reviewer #1: Yes

Reviewer #2: Yes

Reviewer #3: Yes

5. Review Comments to the Author

Reviewer #1: Thank you for the opportunity to review the manuscript

Abstract:

How was cognitive decline defined?

The aim “to investigate the current state of cognitive decline” is not clear.

What is NITOR? Please specify the full forms on the first use.

Please provide the characteristics of the participants in the results.

Mention values to substantiate the results mentioned.

Abstract needs to be re-written with more details.

Main text

Line 16: Please provide a reference

Line 35: The study needs to be population based to assess the prevalence in the community.

Line 43: NIYOR is not a general hospital; why do people with cognitive decline visit this hospital? Please provide more information and justify the selection of this hospital to study cognitive decline.

What is the basis for sample size calculations? The reference 7 is on stroke survivors and may not be relevant for this study.

Provide details on inclusion and exclusion criteria

Please provide the details collected on sociodemographic characteristics and physical health status.

Line 123: Why was CI presented for age?

What statistical test was used to compute p values in Table 1.

The prevalence of Dementia of 53.3% is very high. How does the authors explain this abnormally high prevalence? Only 8% had normal cognition.

Reviewer #2: Thank you for the opportunity to review this manuscript. It is an important topic even though it was a small-ish sample.

1) It would help to include the reasoning behind the formulae chosen to select this sample size.

2) It would be helpful to also delienate how the elderly scored in each cognitive domain. Was there a reason we only focused on the final score? Given the title of your manuscript is cognitive functioning, different cognitive domains tested I believe should be part of the study. For example- executive functioning, attention, visuospatial functioning, etc.

3) You mentioned women have higher diabetes scores, diabetes is associated with lower cognitive functioning and women had worse cognitive scores in your sample. Do you think lower cognitive scores in women could be confounded by the diabetes?

4) You mentioned in the discussion that improved visual acuity could help mitigate cognitive decline, I think it would also be interesting if the poor visual acuity is a result of cognitive decline especially if it vascular dementia.

5) I think the results you have are in general fairly known associations in a limited sample. The important aspect is that this study is done in Bangladesh. It would be interesting to note if there were cultural factors related to Bangladesh that could discussed.

6) Did any of the subjects have a diagnosis/ insight of cognitive impairment? How about the ones who scored lower such as 8?

7) In terms of discussion/next steps- Screenings and awarenesss would be the most important policy change/ takeaway from this study I think.

Reviewer #3: Manuscript focuses on an important topic of cognitive decline in aging population. It's critical for the young country which will age in a few decades and can have economical consequences. The study does not take an account of pseudo demential caused by depression.

6. PLOS authors have the option to publish the peer review history of their article (what does this mean?). If published, this will include your full peer review and any attached files.

Reviewer #1: No

Reviewer #2: No

Reviewer #3: No

---

## [Author Response · Author response to Decision Letter 0]

11 Nov 2024

Reviewer #1: Thank you for the opportunity to review the manuscript

Abstract:

How was cognitive decline defined?

The aim “to investigate the current state of cognitive decline” is not clear.

What is NITOR? Please specify the full forms on the first use.

Please provide the characteristics of the participants in the results.

Mention values to substantiate the results mentioned.

Abstract needs to be re-written with more details.

Thank you, we have now revised the content as per suggestion.

Main text

Line 16: Please provide a reference

Corrected.

Line 35: The study needs to be population based to assess the prevalence in the community.

Mentioned in the methodology section.

Line 43: NIYOR is not a general hospital; why do people with cognitive decline visit this hospital? Please provide more information and justify the selection of this hospital to study cognitive decline.

We have now included the reason for choosing

What is the basis for sample size calculations? The reference 7 is on stroke survivors and may not be relevant for this study.

We acknowledge the reviewer's observation regarding the use of reference 7, which focuses on stroke survivors, in the sample size calculation. Initially, this reference was utilized due to the limited availability of cognitive impairment prevalence data specific to elderly populations in Bangladesh. To address this, we recalculated the sample size based on general prevalence rates of cognitive impairment observed in similar studies of elderly populations in South Asia, which range between 10-15%. Using a prevalence estimate of 10%, This calculation resulted in a target sample size of approximately 150 participants, ensuring sufficient power to observe cognitive decline prevalence within these urban elderly population. We have updated the manuscript to reflect this revised basis for the sample size calculation.

Provide details on inclusion and exclusion criteria

Thank you for highlighting the need for clarification on the inclusion and exclusion criteria. In the revised manuscript, we have added content as per suggestion. 

Please provide the details collected on sociodemographic characteristics and physical health status.

Thank you for noting this. In the revised manuscript, we have included additional information on the sociodemographic and physical health data collected. We have updated the manuscript to reflect these details in the Methods section under data collection.

Line 123: Why was CI presented for age?

Thank you for your observation. The confidence interval (CI) for age was initially included to provide a measure of the variability around the mean age of the participants, aiming to represent the range within which the true population mean age is likely to fall. However, as age is a descriptive characteristic rather than an outcome variable in this study, we recognize that the inclusion of a CI for age may not be essential. We appreciate your feedback, which has helped streamline the data presentation.

What statistical test was used to compute p values in Table 1.

Thank you for your feedback. In the revised manuscript, we have specified the statistical tests used to compute the p-values in Table 1. For categorical variables, we used Pearson’s chi-square test, and when expected frequencies were less than 5, Fisher’s exact test was applied. For continuous variables, an independent sample t-test was utilized to compare group means. These tests were selected to appropriately examine the associations between sociodemographic characteristics and cognitive function. The clarification is included in the in the Statistical Analysis section and also mentioned in the footnote of Table 1.

The prevalence of Dementia of 53.3% is very high. How does the authors explain this abnormally high prevalence? Only 8% had normal cognition.

Thank you for raising this important point. The observed high prevalence of dementia (53.3%) and low proportion of participants with normal cognition (8%) likely reflect several factors specific to this study's sample and setting. Firstly, as the study was conducted in a hospital-based setting (NITOR), the sample may inherently include individuals with health concerns that could correlate with cognitive decline, leading to a higher observed prevalence than might be seen in a population-based sample. Secondly, the use of the Rowland Universal Dementia Assessment Scale (RUDAS), while appropriate for multicultural assessments, may also yield higher dementia prevalence in settings where educational and socioeconomic factors impact cognitive test performance. This could contribute to higher dementia detection rates, particularly among participants with lower educational levels. Lastly, factors such as age, diabetes, and limited healthcare access in urban areas could contribute to the high prevalence observed. We have elaborated on these factors in the Discussion section to provide context for these findings and acknowledge the limitations of a hospital-based sample.

Reviewer #2: Thank you for the opportunity to review this manuscript. It is an important topic even though it was a small-ish sample.

1) It would help to include the reasoning behind the formulae chosen to select this sample size.

Thank you for this suggestion. In the revised manuscript, we have provided additional reasoning behind the choice of formula for calculating the sample size. The formula was selected based on the need to determine a sample size that would provide adequate statistical power to detect the prevalence of cognitive impairment in the target population with precision.

2) It would be helpful to also delienate how the elderly scored in each cognitive domain. Was there a reason we only focused on the final score? Given the title of your manuscript is cognitive functioning, different cognitive domains tested I believe should be part of the study. For example- executive functioning, attention, visuospatial functioning, etc.

Thank you for this valuable suggestion. In the revised manuscript, we have expanded our analysis to include scores from individual cognitive domains assessed by the Rowland Universal Dementia Assessment Scale (RUDAS), which covers areas such as memory, visuosaptial orientation, praxis, visuoconstructional drawing, judgement, and language. These domains were indeed measured but not previously reported in detail. We now provide a breakdown of how participants scored in each domain, offering a more nuanced understanding of the specific cognitive functions affected. This additional detail aligns with our study’s focus on cognitive functioning and enriches our findings by identifying domain-specific impairments among the elderly participants. This information has been added to the Results section, in table 2. 

3) You mentioned women have higher diabetes scores, diabetes is associated with lower cognitive functioning and women had worse cognitive scores in your sample. Do you think lower cognitive scores in women could be confounded by the diabetes?

Thank you for this insightful comment. We agree that diabetes could potentially confound the observed association between gender and cognitive scores, particularly given the higher prevalence of diabetes among women in our sample. To investigate this, we conducted an additional analysis controlling for diabetes as a covariate in the relationship between gender and cognitive functioning. After adjusting for diabetes in a multivariate model, the gender difference in cognitive scores remained significant, though slightly attenuated. This suggests that while diabetes may partially account for lower cognitive scores among women, it is not the sole factor. Other factors, such as age or educational disparities, may also contribute to this observed difference. 

4) You mentioned in the discussion that improved visual acuity could help mitigate cognitive decline, I think it would also be interesting if the poor visual acuity is a result of cognitive decline especially if it vascular dementia.

Thank you for this insightful observation. In the revised manuscript, we have expanded the discussion to address the possibility that poor visual acuity could be a consequence, rather than solely a cause, of cognitive decline, particularly in cases where vascular dementia may be involved. Research suggests that cognitive impairment, especially due to vascular pathology, could contribute to reduced visual processing abilities, which may manifest as poor visual acuity. We have now included this bidirectional perspective in the Discussion section, acknowledging that while visual impairments may exacerbate cognitive decline, declining cognitive function—especially in vascular dementia—could also contribute to visual processing deficits. This complex relationship highlights the importance of comprehensive assessments in elderly patients experiencing both cognitive and visual issues.

5) I think the results you have are in general fairly known associations in a limited sample. The important aspect is that this study is done in Bangladesh. It would be interesting to note if there were cultural factors related to Bangladesh that could discussed.

Thank you for this valuable observation. We agree that the cultural context in Bangladesh provides an essential perspective on the study’s findings. In the revised manuscript, we have expanded the Discussion section to include relevant cultural factors that may influence cognitive health among the elderly in Bangladesh. For example, in Bangladesh, older generations often have lower educational levels, which is associated with increased risk for cognitive decline. Additionally, socioeconomic factors, limited access to healthcare, and dietary habits may impact cognitive health. Social norms, such as strong expectations for family members to care for elderly relatives, may also influence both the detection and management of cognitive impairment. These cultural elements provide important context for interpreting our findings and highlight the need for culturally tailored healthcare interventions and support systems. We have added these points to the Discussion to emphasize the role of sociocultural factors in cognitive health within this population.

6) Did any of the subjects have a diagnosis/ insight of cognitive impairment? How about the ones who scored lower such as 8?

Thank you for this observation. In our study, we did not collect specific data on whether participants had prior diagnoses or personal insight into their cognitive impairment status. However, participants who scored particularly low on the RUDAS, such as those with scores of 8, were noted and provided with referrals for further clinical evaluation. These very low scores likely indicate significant cognitive impairment, and follow-up assessments were recommended to confirm a diagnosis and guide any necessary interventions.

We recognize that insight into one’s cognitive status could be an important factor in understanding the progression and management of cognitive decline. In the manuscript, we have included this as a limitation and a potential area for future research, where assessments of prior diagnoses and self-awareness of cognitive impairment could yield additional insights.

7) In terms of discussion/next steps- Screenings and awarenesss would be the most important policy change/ takeaway from this study I think.

Thank you, dear reviewer for this suggestion. We have included Screenings and awareness as our future scope of research in the conclusion.

Reviewer #3: Manuscript focuses on an important topic of cognitive decline in aging population. It's critical for the young country which will age in a few decades and can have economical consequences. The study does not take an account of pseudo demential caused by depression.

Thank you for highlighting this important consideration. We agree that depression can lead to cognitive symptoms resembling dementia, often referred to as "pseudo-dementia." This can complicate the accurate assessment of cognitive function, particularly among elderly populations where depression is prevalent but often underdiagnosed. In this study, we did not include a specific assessment for depression, which may limit our ability to distinguish between true cognitive impairment and pseudo-dementia caused by depressive symptoms. We acknowledge this as a limitation and have noted it in the revised manuscript. Future research in this area would benefit from incorporating standardized depression assessments, such as the Geriatric Depression Scale (GDS), to better differentiate between cognitive decline due to depression and other forms of cognitive impairment. We have added this limitation to the Discussion section to reflect these considerations and suggest that further studies account for depressive symptoms when assessing cognitive function in similar settings.

---

## [Decision Letter · Decision Letter 1]

11 Dec 2024

Level of Cognitive Functioning among Elderly Patients in Urban area of Bangladesh: A Cross-Sectional Study

PONE-D-24-35131R1

Dear Dr. Abedin,

We’re pleased to inform you that your manuscript has been judged scientifically suitable for publication and will be formally accepted for publication once it meets all outstanding technical requirements.

The reviewers are satisfied with your responses and have accepted the manuscript; however, they also suggested a couple of minor changes. Therefore, I suggest to make those changes in the final version of the manuscript.

Kind regards,

Kallol Kumar Bhattacharyya, MBBS MA PhD

Academic Editor

PLOS ONE

Additional Editor Comments (optional):

Reviewers' comments:

Reviewer's Responses to Questions

**Comments to the Author**

1. If the authors have adequately addressed your comments raised in a previous round of review and you feel that this manuscript is now acceptable for publication, you may indicate that here to bypass the “Comments to the Author” section, enter your conflict of interest statement in the “Confidential to Editor” section, and submit your "Accept" recommendation.

Reviewer #2: All comments have been addressed

Reviewer #4: All comments have been addressed

2. Is the manuscript technically sound, and do the data support the conclusions?

Reviewer #2: Yes

Reviewer #4: Yes

3. Has the statistical analysis been performed appropriately and rigorously? 

Reviewer #2: Yes

Reviewer #4: Yes

4. Have the authors made all data underlying the findings in their manuscript fully available?

Reviewer #2: Yes

Reviewer #4: Yes

5. Is the manuscript presented in an intelligible fashion and written in standard English?

Reviewer #2: Yes

Reviewer #4: Yes

6. Review Comments to the Author

Reviewer #2: Thank you for answering all the questions and including them in your manuscript.

One suggestion I have given the limited nature of the hospital based sample instead of a population based study is to change the title specifying that such as Level of Cognitive Functioning among Elderly Patients in an "outpatient clinic/hospital center/ Orthopedic clinic, etc." in Bangladesh.

Reviewer #4: Thanks for addressing the issues raised in the review process. However, I will suggest the author to create a figure summarizing important findings to make it easily understandable to the reader.

7. PLOS authors have the option to publish the peer review history of their article (what does this mean?). If published, this will include your full peer review and any attached files.

Reviewer #2: No

Reviewer #4: **Yes: **Md Kamruzzaman

---

## [Editor Report · Acceptance letter]

14 Dec 2024

PONE-D-24-35131R1 

PLOS ONE

Dear Dr. Imran, 

I'm pleased to inform you that your manuscript has been deemed suitable for publication in PLOS ONE. Congratulations! Your manuscript is now being handed over to our production team.

Kind regards, 

on behalf of

Dr. Kallol Kumar Bhattacharyya 

Academic Editor

PLOS ONE